# Comparison of Estimated Wild Giant Anteater (*Myrmecophaga tridactyla* Linnaeus, 1758) Diets with Commercial Diets for Insectivores: Implications for Anteater Health

**DOI:** 10.3390/ani13233606

**Published:** 2023-11-22

**Authors:** Heidi Bissell, Mario H. Alves, Débora R. Yogui, Margarita Woc Colburn, Arnaud L. J. Desbiez

**Affiliations:** 1Busch Gardens Tampa Bay, 3605 E Bougainvillea Ave, Tampa, FL 33612, USA; heidi.bissell@disney.com; 2Instituto de Conservação de Animais Silvestres (ICAS), Rua Licuala 622, Campo Grande 79046-150, Mato Grosso do Sul, Brazil; adesbiez@hotmail.com (A.L.J.D.); 3Nashville Zoo, 3777 Nolensville Pike, Nashville, TN 37211, USA; 4Arizona Center for Nature Conservation, Phoenix Zoo, 455 N. Galvin Parkway, Phoenix, AZ 85008, USA; 5Instituto de Pesquisas Ecológicas (IPÊ), Rodovia Dom Pedro I, km 47, Nazare Paulista 12960-000, Sao Paulo, Brazil; 6Royal Zoological Society of Scotland (RZSS), Murrayfield, Edinburgh EH12 6TS, UK

**Keywords:** nutrition, Pilosa, wildlife, Xenarthra

## Abstract

**Simple Summary:**

Anteaters are insectivores, consuming diets consisting primarily of termites and ants in the wild; therefore, feeding them under human care has been a challenge and frequently leads to nutritional disorders. Through analysis of free-ranging giant anteaters’ stomach content, the findings of this study provide valuable information for optimizing anteater nutrition in zoological institutions and rehabilitation centers causing a significant impact on the welfare of the species.

**Abstract:**

Using the stomach contents of 24 wild road-killed giant anteaters as proxies for their diet, we found that estimated wild diets were much lower in calcium (about ten times less) and phosphorus compared with manufactured feeds commonly fed to anteaters under human care. Indicators of soil consumption, such as ash, iron, and manganese were higher in the stomach contents than in either wild termites or manufactured diets, likely due to high levels of soil ingestion during feeding in the wild. Protein and fat levels in insects, stomach contents, and commercial diets all met carnivore recommendations. Both giant anteaters and tamanduas in managed care often develop hypercalcemia, perhaps because these taxa have an enhanced ability to retain calcium allowing them to survive on such low calcium diets. Results from this study indicate that, for anteaters in managed care, it is important to keep dietary calcium and vitamins D and K within recommended levels to prevent nutritional diseases such as hyper- and hypocalcemia and vitamin K deficiency.

## 1. Introduction

Anteaters are part of a unique ancient lineage of mammals, the Xenarthrans, that includes anteaters, armadillos, and sloths. This lineage has many unique characteristics and is not closely related to other groups of mammals [1]. Anteaters are insectivores, consuming diets consisting primarily of termites and ants in the wild [2,3]. Ants and termites are difficult for most zoos to obtain, manage, and distribute. In managed care anteaters are often fed either commercial insectivore diets or gruels made of supplemented meat and domestic cat or dog food. Anteaters under human care, including both giant anteaters (*Myrmecophaga tridactyla*) and southern tamanduas (*Tamandua tetradactyla*), as well as several other members of Xenarthra are frequently diagnosed with hypercalcemia, that can lead to vertebral hyperostosis, weakness, hypotonia, hyporeflexia, and cardiovascular effects such arrhythmias and hypertension [4,5,6,7]. This has been attributed to excessive levels of dietary vitamin A, vitamin D, and/or calcium.

Excessive dietary calcium interferes with the absorption of minerals such as P, Zn, Fe, Cu, and Mn as well as vitamin K absorption [7,8]. Giant anteaters under human care have exhibited blood clotting disorders associated with vitamin K deficiency despite seemingly sufficient dietary levels [4,5,9,10,11]. Anteaters, particularly the southern tamandua, are routinely supplemented (either via a commercial diet or separately) with vitamin K to prevent clotting disorders [12].

In this study, we compare the nutrient contents of the stomachs of 24 wild giant anteaters with the previous literature about tamanduas and insect nutrient composition and compare this with several available commercial insectivore diets. This is the first study published with nutritional analysis of free-ranging giant anteaters’ stomach contents. We use our results to evaluate current zoo feeding recommendations, zoo commercial insectivore diets, and to evaluate possible causes of some nutritional concerns seen in anteaters.

## 2. Materials and Methods

Stomach contents were collected from 24 road-killed giant anteaters along public roadways in Mato Grosso do Sul, Brazil (Figure 1). This study was performed under license No. 53798 (Chico Mendes Institute for Biodiversity Conservation) that granted permission to collect and store biological samples. The age of the carcasses was generally less than a few hours old, as determined by carcass condition. The date, GPS coordinates of the carcass, name of the road section, and sex of the animal were recorded. In the field, the stomach was separated by tying off the esophageal and pyloric sphincters with string before removing it from the animal.

The stomach contents were weighed and frozen at −20 °C for further analysis at CBO Lab, Valinhos in Sao Paulo, Brazil. Analyses included dry matter as well as crude protein, fat (ether extract), gross energy, ash, calcium, phosphorus, magnesium, iron, and manganese. The levels of these nutrients were evaluated on a dry matter (% DM) as well as on an energy basis (g or mg per 1000 kcal) because of the high rate of soil inclusion and because carnivore diets are most commonly and best evaluated on an energy basis. We calculated the fat:protein kcal ratio as (9 × fat)/(4 × protein) as well as the Ca:P ratio. These samples enable us to examine the nutrient profile or value of the environment’s resources as sampled by the anteaters themselves. Unfortunately, we were not able to test stomach contents for vitamin levels.

Finally, we compared the nutrient composition of the stomach contents to previously reported values for tamandua stomachs and termites [13,14,15,16] as well as several commercially available insectivore diets from the US and Brazil using analysis of variance (ANOVA). Commercial diets were analyzed at Dairy One Laboratories in Ithaca, NY, for proximate composition and minerals, gross energy, neutral detergent and acid detergent fiber, and acid insoluble ash using Association of Official Analytical Chemists-recognized wet chemistry procedures [17] or, in the case of the Brazilian feed, using the manufacturer’s printed specifications.

In figures, analyses were grouped by feed type and shown as means ± standard deviation.

## 3. Results

The nutrient composition of the material found in the 24 stomachs of wild anteaters from this study are summarized on a dry matter basis (Table 1) and an energy basis (Table 2). More details about the results of each sample are provided in the Appendix A.

### Comparison with Manufactured Feeds

Figure 2 and Figure 3 compare the results from this study to published estimates of termite composition [13,14,15,16] and analyses of several commonly manufactured zoo feeds.

On a dry matter basis (Figure 2), the stomach contents of wild anteaters were higher in ash, iron, and manganese than either termites or manufactured diets (*p* < 0.05). Commercial diets were much higher in calcium and phosphorus (*p* < 0.001) and had a higher Ca:P ratio than either insects or the stomach contents. The fat:protein ratio was lower in the stomach contents than in insects or manufactured diets (*p* < 0.1).

On an energy basis (Figure 3), the stomach contents were higher in protein (*p* < 0.05), ash (*p* < 0.001), iron (*p* < 0.001), manganese (*p* < 0.001), and had a higher fat:protein ratio (*p* < 0.001) than commercial diets. Commercial diets were much higher in Ca (*p* < 0.001), P (*p* < 0.005), and Ca:P ratio (*p* < 0.001) than either the stomach contents or insects. The nutrient composition of the material found in the 24 stomachs of wild anteaters from this study are summarized on a dry matter basis (Table 1) and an energy basis (Table 2). More details about the result of each sample are provided in the Appendix A.

## 4. Discussion

### 4.1. Macronutrients (Protein, Fat, Ash)

Giant anteater stomach contents were extremely high in ash (61.8%), likely due to a high level of soil inclusion in the diet consumed. This result is much higher than the data previously reported for southern tamandua (13.8%) [14]. The edentulous giant anteater ingests insects using a slender, elongated sticky tongue [18] during which soil is ingested [19]

Fat content was extremely variable in both stomach contents and insects, likely due to the high level of variability in insect composition. Insect nutrient composition was obtained from the literature and included a variety of life stages and sampling techniques, which could greatly increase variability. In general, younger stages of insects (larvae, grubs) tend to be fattier than more mature adult stages. Fat content was considerably lower in the stomach contents than in insects or commercial insectivore diets, however, indicating that anteaters may be primarily consuming older insect life stages with lower fat composition.

Macrominerals: Levels of calcium in the giant anteater stomachs sampled in this study (0.4 g/Mcal, 0.1%DM), in terms of dry matter, were similar to levels reported in the stomach contents of southern tamanduas (0.24 g/Mcal (0.11% DM)) [14]. Both values, in terms of dry matter, are lower than the calcium content reported for termites themselves (0.20 g/Mcal, 0.26 ± 0.23% DM.) [14]. Levels of calcium in the giant anteater stomachs are much lower (about ten times less) than levels found in manufactured diets (3.6 ± 0.5 g Ca/Mcal or 1.58 ± 0.19% DM). Levels of calcium in all insects and all stomach contents were lower than recommended for dogs (1 g/Mcal) and cats (0.72 g/Mcal) [20].

Observations of wild giant anteaters have never indicated that anteaters are attracted to sources of salt or minerals. During more than 350 h of field observations with giant anteaters in areas where mineral salt mixtures were available to local cattle, the anteaters were not observed exhibiting any calcium or phosphorus-seeking behaviors [21].

Typical mammalian calcium requirements for non-lactating animals are in the range of 0.2–1.0% of diet dry matter, with safe upper limits of 2.0% [8]. Most commercial feeds for animals contain 1.0–1.8% calcium. Both giant anteater (this study) and southern tamandua [14] anteater stomach contents appear to contain calcium levels approximately an order of magnitude lower than this.

Furthermore, the Ca:P ratio found in giant anteater stomach content (0.5 ± 0.4) is well below 1.0. In other mammals, this would interfere with adequate calcium absorption, further increasing the risk of calcium deficiency [8,22,23]. Yet, anteaters are routinely diagnosed with conditions reflective of excessive calcium [4,24,25].

### 4.2. Calcium and Phosphorus Metabolism and Current Recommendations

Xenarthrans may have a specialized calcium metabolism that enables them to survive on such low calcium diets in the wild without developing hypocalcemia. Lines of evidence for this come from several different directions. First, zoo anteaters commonly experience hypercalcemia on diets containing “typical” levels of calcium, phosphorus, and vitamin D required by and fed to many other mammals. Anteaters (family Myrmecophagidae), particularly the southern tamandua, are prone to this condition, which can lead to calcification of joints, arteries, and organs and eventually multi-organ failure [4,7,24]. In most mammals, calcium and phosphorus compete for absorption. When diets contain higher phosphorus than calcium, bone loss can result [23]. Therefore, commercial animal feeds are typically manufactured to have slightly higher calcium than phosphorus. A Ca:P ratio of 1–1.8 is generally recommended [20]. However, both insects and the wild anteater stomach contents had Ca:P ratios far below 1, indicating that anteaters may be able to absorb sufficient calcium even with an inverted Ca:P ratio. Previous studies have attributed the hypercalcemia to excesses of vitamins A or D as well as calcium [4,25]. We were unable to evaluate vitamins in this study, but they may also play a role in increasing the likelihood of hypercalcemia.

A second line of evidence can be seen when we estimate intake levels. We typically estimate energy requirements of giant anteaters using marsupial energy calculations because both groups have a similarly low basal metabolic rate. Using this method, a 35 kg adult giant anteater would require approximately 1140 kcal/day [26]. With an average of 712 kcal/kg in the stomach contents as sampled, the animal would need to eat approximately 1.6 kg of food each day. The weight of stomach contents in the sampled anteaters ranged from 145 g to over 2.2 kg (mean ± SD: 828 g ± 700 g), so it seems very reasonable that an anteater can meet its energy requirements on the current diet. However, when consuming 1.6 kg, or even 2.2 kg, of food, the animal would consume less than 1 mg of calcium a day. For comparison, a 35 kg domestic dog requires 1900 mg of calcium per day [20]. An anteater would have to consume over 9.5 kg, nearly a third of its body weight, each day to obtain that much calcium on the diet we estimated from the stomach contents. That is exceedingly unlikely, and they have no other known sources of calcium, so we must assume that they are able to survive on much lower levels of calcium in their diet.

A third intriguing line of evidence comes from studies of other xenarthran milks. Armadillo milk is extremely high in both protein and calcium compared with other mammalian milks. A previous study hypothesized that the high protein content is needed to suspend the calcium in solution (calcium is otherwise insoluble in water), and that the elevated calcium levels are needed to help build the bony shell of the growing armadillo [27]. Yet, other xenarthrans also have extremely high protein milk. In fact, anteaters have the highest protein energy of any mammalian milk analyzed to date. The author of that study hypothesized that these high protein/calcium milks might be a trait shared among xenarthrans [27]. Although giant anteaters do not have the bony carapace of armadillos, their diet appears to be extremely low in calcium. Consumption of high calcium milk coupled with strong calcium absorption mechanisms could be essential to helping a young anteater develop calcium stores that enable them to grow, even on a low calcium diet.

And, finally, the fourth line of evidence comes from the field: it is unlikely that hypocalcemia is a concern for wild populations. Giant anteaters in a nearby study area within the same state have been monitored and tracked using GPS collars and have been observed successfully reproducing offspring, indicating that the wild population can survive and produce healthy offspring at these low dietary Ca and P levels [28]. Additionally, mean total calcium blood levels reported for free-ranging anteaters from the same state were under normal levels (8.82 mg/dL) [29] and similar to mean values for zoo anteaters (9.3 mg/dL) [30], validating the notion that even when consuming low calcium diets they are able to maintain normal calcium levels.

Care should be exercised when interpreting data from estimated wild diets. First, these wild diet figures are merely estimates, with errors due to sampling (stomach contents reflect approximately one meal vs. intake over time) and digestion (contents likely experienced at least some digestion and passage, which is often not uniform across all food types). Second, wild diets are not necessarily “ideal” diets. They merely represent what the animal was able to obtain in that location on that day and have the potential to be unbalanced or insufficient. Finally, we do not know the quantities of intake. Diets with low proportions of important nutrients can still provide enough of those nutrients if the animal is able to consume large enough amounts. However, based on calculations above, even doubling the current dietary intake would not meet calcium requirements for domestic carnivores. Balance studies of calcium intake and excretion on different levels of dietary calcium would help us resolve whether anteaters have enhanced calcium retention compared with other mammals.

To address the growing concerns with hypercalcemia in xenarthrans, the Pangolin, Aardvark, and Xenarthra Taxon Advisory Group (PAX TAG) of the Association of Zoos and Aquariums (AZA) proposed dietary recommendations of 11–20 mcg (440–800 IU)/1000 kcal of diet of vitamin D (equivalent to 1320–2400 IU/kg DM) and maintaining calcium levels at 0.5–1.3% (max) of diet dry matter. To assess the status of animals in managed care, target serum vitamin D levels are recommended to be 25–100 nmol/L and serum calcium levels 8–11 mg/dL for tamanduas and 7.8–10 mg/dL for giant anteaters [31]. If serum values are outside of these ranges, or if calcium deposits are seen on radiographs, the diet should be adjusted within the recommended ranges to produce the target serum values.

### 4.3. Vitamin K Metabolism and Current Recommendations

Intriguingly, vitamin K and calcium metabolism are closely intertwined. Pigs fed high Ca diets (a 3:1 Ca:P ratio) without supplemental vitamin K exhibited hemorrhage and increased clotting times, which was not observed in pigs fed diets with a 2:1 or 1:1 ratio despite the same levels of vitamin K. Supplementing 5 mg/kg feed of menadione (vitamin K) normalized the pigs’ clotting parameters, even on the high Ca diets [21]. It is quite possible that excessive Ca interferes with vitamin K absorption or metabolism and leads to an increased vitamin K requirement. This helps explain why anteaters have such a uniquely high vitamin K requirement in managed care—the relatively “high” levels of dietary calcium (in comparison to apparent wild intake levels) may interfere with K metabolism, necessitating supplementation of K. However, vitamin K supplementation above requirements are known to reduce urinary calcium losses (i.e., increase calcium retention) in other rodents and primates, including humans [32,33,34]. If too much supplemental vitamin K is provided (through well-meant efforts to prevent vitamin K deficiency and clotting disorders), this could actually lead to *increased* calcium retention and hypercalcemia in anteaters in managed care. Therefore, vitamin K levels as well as Ca and vitamin D levels, should be carefully kept within recommended ranges. The *Merck Veterinary Manual* recommends providing 5 mg of menadione sodium bisulfite (a form of vitamin K) per kg of dry diet [35], which is higher than the 1–1.6 mg/kg diet recommended by the NRC for dogs and cats [20]. The target level for dietary vitamin K may ultimately depend on dietary Ca levels, but keeping K within the range of 1–5 mg/kg diet is likely to provide sufficient amounts of K to prevent clotting disorders while not over-supplementing and causing additional calcium retention, especially if dietary Ca levels can also be reduced. Understanding anteaters’ seemingly unique relationship with calcium and vitamin K is an intriguing and important area for future work.

## 5. Conclusions

Estimates of anteater diets from their stomach contents reveal that wild giant anteaters consume diets that are extremely low in calcium and phosphorus levels and have an inverted Ca:P ratio compared with the requirements of other mammals. However, it appears that anteaters have mechanisms to enhance absorption and/or retention of calcium. Therefore, for anteaters in managed care, it is important to keep dietary calcium and vitamins D and K within recommended levels in order to prevent nutritional diseases such as hypercalcemia, hypocalcemia, and vitamin K deficiency. Further research is needed to understand the relationship between optimal dietary levels of vitamin D, calcium, and vitamin K in giant anteaters, southern tamanduas, and other xenarthrans.

## Figures and Tables

**Figure 1 animals-13-03606-f001:**
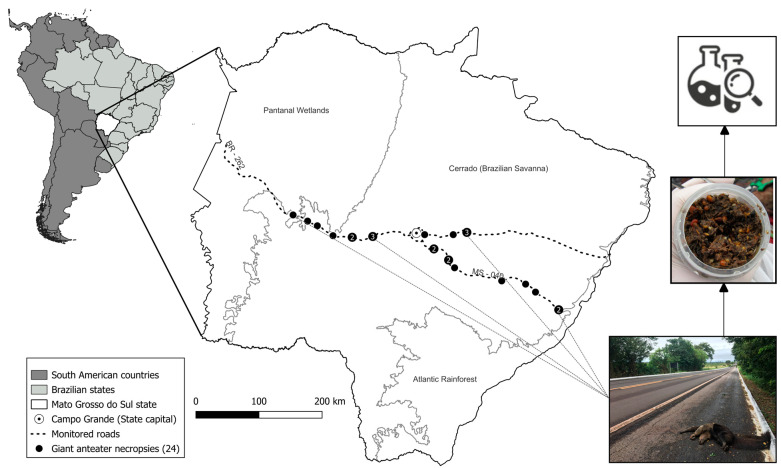
Map showing where road-killed giant anteaters were necropsied in the Mato Grosso do Sul state, Brazil. The specimens found are represented by black dots; numbered when two or more are located nearby. All the necropsies were near the national road BR-262 and the state road MS-040. Stomach samples were collected in the field and sent for analyses.

**Figure 2 animals-13-03606-f002:**
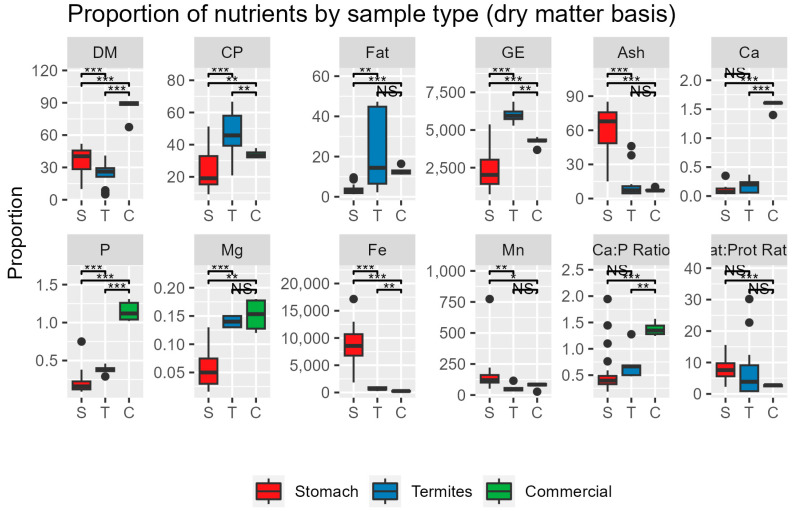
DM is expressed as % of sample. Protein, fat, ash, Ca, P, Mg are expressed as % of dry matter, Fe and Mn are expressed as ppm of dry matter; energy is expressed as kcal/kg. *** *p* < 0.001, ** *p* < 0.01, * *p* < 0.05. NS = Not significant.

**Figure 3 animals-13-03606-f003:**
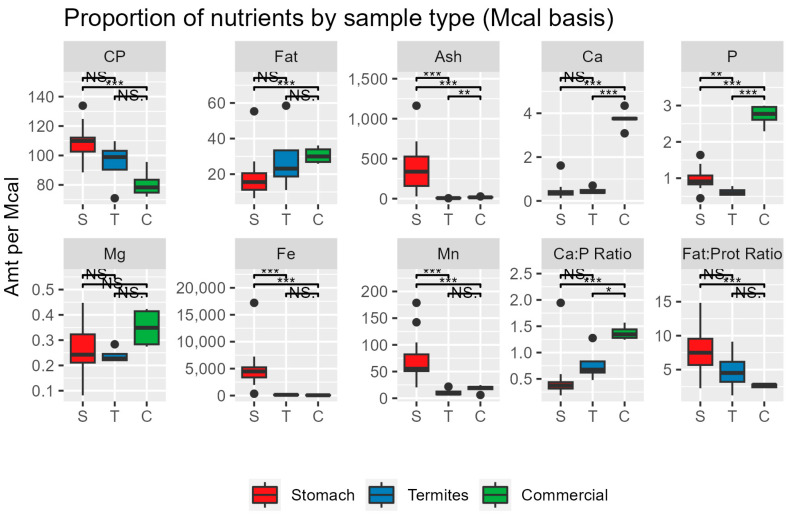
Values are expressed as amount per Mcal. Protein, Fat, Ash, Ca, P, Mg are expressed as g/Mcal, Fe and Mn are expressed as mg/Mcal of dry matter. *** *p* < 0.001, ** *p* < 0.01, * *p* < 0.05. NS = Not significant.

**Table 1 animals-13-03606-t001:** Analyses on a dry matter basis of the stomach contents of 24 giant anteaters compared with selected commercial insectivore diets, literature values for termites (all life stages), and NRC recommendations for dogs and cats. Values ± standard deviation.

Analysis	Stomach Contents (This Study)	Termites	Commercial Diets	NRC Canine Recommendations	NRC Feline Recommendations
Dry Matter (%)	36.3 ± 2.5	22.7 ± 2.5	85.3 ± 4.5	-	-
Crude Protein (%)	24.1 ± 2.5	45.7 ± 3.04	34 ± 1.16	10	20
Crude Fat (%)	3.5 ± 0.5	22.2 ± 5.4	12.9 ± 0.1	5.5	9
Gross Energy (kcal/kg)	2300 ± 260	6000 ± 330	4230 ± 145	-	-
Ash (%)	61.8 ± 3.8	11.0 ± 2.9	7.8 ± 1.4	-	-
Calcium (%)	0.1 ± 0.01	0.17 ± 0.04	1.6 ± 0.04	0.4	0.29
Phosphorus (%)	0.21 ± 0.03	0.38 ± 0.03	1.2 ± 0.06	0.3	0.26
Magnesium (%)	0.1 ± 0	0.14 ± 0.0	0.15 ± 0.02	0.06	0.04
Iron (ppm)	8860 ± 730	690 ± 100	240 ± 34	30	80
Manganese (ppm)	158 ± 32.2	57 ± 15	74 ± 12	4.8	4.8
Ca:P	0.5 ± 0.08	0.7 ± 0.14	1.4 ± 0.06	1.3	1.1
Fat:Protein	7.9 ± 0.7	7.5 ± 2.6	2.67 ± 0.13	0.55	0.45

**Table 2 animals-13-03606-t002:** Analyses on an energy basis of the stomach contents of 24 giant anteaters compared with selected commercial insectivore diets, literature values for termites (all life stages), and NRC recommendations for dogs and cats. Values ± standard deviation.

Analysis	Stomach Contents (This Study)	Termites	Commercial Diets	NRC Canine Recommendations	NRC Feline Recommendations
Crude Protein (g/Mcal)	109 ± 2.4	94.6 ± 8.3	80.8 ± 4.16	25	50
Crude Fat (g/Mcal)	17.2 ± 2.2	29.0 ± 10.3	30.6 ± 2.0	13.8	22.5
Ash (g/Mcal)	379 ± 59	6.7 ± 0.46	18.7 ± 2.3	-	-
Calcium (g/Mcal)	0.4 ± 0.07	0.46 ± 0.08	3.7 ± 0.2	1.0	0.72
Phosphorus (g/Mcal)	0.96 ± 0.05	0.62 ± 0.06	2.7 ± 0.13	0.75	0.64
Magnesium (mg/Mcal)	0.27 ± 0.02	0.24 ± 0.02	0.35 ± 0.04	150	100
Iron (mg/Mcal)	4830 ± 707	124 ± 34	56 ± 17	7.5	20
Manganese (mg/Mcal)	71 ± 8.01	11.1 ± 3.6	56.4 ± 7.44	4.5	1.2
Ca:P	0.45 ± 0.08	0.78 ± 0.17	1.4 ± 0.1	1.3	1.1
Fat:Protein	7.7 ± 0.67	4.8 ± 1.6	2.67 ± 0.13	0.55	0.45

## Data Availability

Data are contained within the article and Appendix A.

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
