# Peer review of "Comparison of Estimated Wild Giant Anteater (Myrmecopahaga tridactyla Linnaeus, 1758) Diets with Commercial Diets for Insectivores: Implications for Anteater Health"

_animals, 2023, doi:10.3390/ani13233606_

Round 1

Reviewer 1 Report

Comments and Suggestions for Authors

I read this work with great interest, and I am impressed that the authors undertook the difficult task of collecting stomach content from road-killed giant anteaters. In this respect, it is necessary to specify where (in the laboratory or in the field) the material for analysis was collected.

Since the variable 'sample weight' (supplementary table) does not show any variability, I suggest adding a few decimal places e.g. 0.254 kg.

I strongly suggest also adding concentrations for Ca, P and Mg (to supplementary table) because these data may be cited in the future. Similarly, please expand the values for Ca, P and Mg in Table 1.

I also see that it would be necessary to check whether there are sex-related differences for the variables mentioned in supplementary table.

Author Response

COMMENTS FROM REVIEWER 1

Comment 1: [I read this work with great interest, and I am impressed that the authors undertook the difficult task of collecting stomach content from road-killed giant anteaters. In this respect, it is necessary to specify where (in the laboratory or in the field) the material for analysis was collected.]

Response: Thank you for your comments and recommendations. We agree with that. The material was collected in the field during the necropsies, we have added some words in the line 70 to point this out and in the description of Figure 1. The changes are highlighted in the line 70 and in the description of Figure 1.

Comment 2: [Since the variable 'sample weight' (supplementary table) does not show any variability, I suggest adding a few decimal places e.g. 0.254 kg.]

Response: Thank you so much for this suggestion, it helped us to rearrange the supplementary table and let values from both columns “Stomach content weight (Kg)” and “Sample weight (Kg)” with the same number of decimal places.  

Comment 3: [I strongly suggest also adding concentrations for Ca, P and Mg (to supplementary table) because these data may be cited in the future. Similarly, please expand the values for Ca, P and Mg in Table 1.]

Response: Thank you for your comment, we agree that it is very important to share all the raw data through supplementary table allowing future citations and even different analyses by others. However, we have might not understood your recommendation since all the Ca, P and Mg data we have all already provided in both supplementary table and table 1.

Comment 4: [I also see that it would be necessary to check whether there are sex-related differences for the variables mentioned in supplementary table]

Response: Thank you so much for your suggestion, even considering that it would be very interesting to investigate sex-related differences, due to our small sample size (14 males and 10 females), we consider we are not able to generate a reliable data on sex related differences.

Reviewer 2 Report

Comments and Suggestions for Authors

This manuscript provides useful information that has not before been reported on for this species. Results are therefore important to inform captive management of giant anteaters and future work to elaborate on findings here. There is, however, some beneficial details that should be added to improve the overall contribution of the manuscript. There are also several areas in which sentences as written do not make sense and may have been included in error. Below I will indicate specific questions and areas of text I am referring to.

Line 53: Do you mean giant anteaters in human care have exhibited blood clotting disordered? It would be helpful to make your point to include this clarification in the text.

Line 62: The last sentence of the introduction, the line after the comma does not make sense to me. Are there a few words missing? If this is as intended, the point authors are trying to make here is not clear and should be reworded.

Introduction overall: The aim of the study is not clearly conveyed. The point should be made that studies of the wild diet of giant anteaters are lacking and this is the first investigation intending to shed light on this topic. 

Line 64: Was any consideration given for differences in the stomach content of the study animals from giant anteaters that weren't killed on the road? Perhaps these animals were near the road for some behavioral or physiological reason that might impact the results of the study. This possibility should be addressed in the discussion of the paper near where authors nicely describe other reasons that caution should be employed when interpreting wild diet data from stomach contents.

Line 83: Were the reported values for nutrient composition of termites the same as for the analysis of tamandua stomachs? These several papers are referenced together and only one set of values was included for comparison in the paper. Were values from analysis of straight termites identical to that from tamandua stomachs? If not, how did the authors determine which values to use and how were they different?

Discussion overall: The general content of the discussion seems appropriate, but some care should be given to making the concepts clearly grouped into logical paragraphs where the point being made is first introduced and then the supporting information offered below. Then some consideration to the flow would be helpful to the reader. As it is, the discussion is extremely choppy and jumps from one concept to another without clarity around the point the information being given is trying to support. At one point it appears that several lines of evidence were being offered to support a singular point, but this argument was not initially made clear to the reader, causing me to need to go back in the text several times to try to tie everything together logically, when the text itself should have done this.

Line 251: This comment about keeping vitamin K, D, and Ca within recommended ranges is arguably the most important of the paper, and yet it is not clear WHICH recommendations the authors are encouraging readers adhere to. Given the results shown in the paper, it stands to reason that current diet recommendations for this species in human care may not be appropriate and might need adjustment. Or do you mean that diet recommendations are fine as they are, but nutrient values in commercial diets do not follow these recommendations, so animal managers need to be aware? If the former, what should be the impact of this work? Further supporting already existing recommendations or suggesting adjustments that are investigated in more detail in future assessments? This is crucial information and as it stands, this main point of the work is almost lost due to a lack of clarity. Please more clearly articulate here, in the conclusion, and in the abstract WHICH recommendations are being suggested readers be mindful of.

Lines 254-257: These appear not to belong in the discussion section and are instead instructions for writing a discussion section of a paper. Please remove them or clarify.

Comments on the Quality of English Language

The English language used in this paper was able to be understood, but there were a few places in which sentences did not make sense or seemed to be included in error. There are indicated specifically in the comments above. Otherwise, conceptual flow and clarity of text could be greatly improved, but this is not a remark on the quality of the English language used.

Author Response

COMMENTS FROM REVIEWER 2

Comment 1: [This manuscript provides useful information that has not before been reported on for this species. Results are therefore important to inform captive management of giant anteaters and future work to elaborate on findings here. There is, however, some beneficial details that should be added to improve the overall contribution of the manuscript. There are also several areas in which sentences as written do not make sense and may have been included in error. Below I will indicate specific questions and areas of text I am referring to.]

Response: Thank you so much for your review, it will help us to improve the paper in general. We will be answering each point below.

Comment 2: [Line 53: Do you mean giant anteaters in human care have exhibited blood clotting disordered? It would be helpful to make your point to include this clarification in the text.]

Response: Thanks for the comment, and yes, all the cited reports [4, 5, 9-11] were under human care. To answer that suggestion, we have pointed that out in the line 53, the change is highlighted.

Comment 3: [Line 62: The last sentence of the introduction, the line after the comma does not make sense to me. Are there a few words missing? If this is as intended, the point authors are trying to make here is not clear and should be reworded.]

Response: Thank you so much for your comment, there were few words missing, and typing errors. We have rearranged and changed the sentence to: “Results from wild anteater stomach nutrient content are then used to evaluate current recommendations, zoo commercial insectivore diets and to evaluate possible causes of some nutritional concerns seen in anteaters.”

Comment 4: [Introduction overall: The aim of the study is not clearly conveyed. The point should be made that studies of the wild diet of giant anteaters are lacking and this is the first investigation intending to shed light on this topic.]

Response: Thank you so much for pointing this out. We agree and have added a sentence at the end of the introduction highlighting highlighting the originality of the study. We also believe your “comment 3” helped us to improve the last sentence of the introduction to better describe our objectives.

Comment 5: [Line 64: Was any consideration given for differences in the stomach content of the study animals from giant anteaters that weren't killed on the road? Perhaps these animals were near the road for some behavioral or physiological reason that might impact the results of the study. This possibility should be addressed in the discussion of the paper near where authors nicely describe other reasons that caution should be employed when interpreting wild diet data from stomach contents.]

Response: Thank you for your comments. There exists no data on stomach contents analyses from giant anteaters that were not road killed to compare with ours. Our study is the first one to investigate that, and all our samples were collected during road-killed animals’ necropsies. We don’t feel that there is a strong bias between non-road-killed animals and ours. Previous studies with neotropical mammals, including the giant anteater, in the same region, have reported a strong evidence that most of the animals were healthy before being road killed (Navas-Suárez et al., 2022 doi: 10.1016/j.jcpa.2022.06.003). Furthermore, the samples in this study could not have been obtained from living animals. Even if living wild anteaters could be ethically trapped and sacrificed, the delay from the last meal due and stress from trapping would alter the stomach contents to a much greater extent than in the samples from road-killed animals obtained in this study. The methods used in this study are the best methods to obtain stomach samples from wild anteaters.

We truly believe that the paragraph from line 223 to line 232 contains all the caution recommendations that are important when interpreting data from this study.

Comment 6: [Line 83: Were the reported values for nutrient composition of termites the same as for the analysis of tamandua stomachs? These several papers are referenced together and only one set of values was included for comparison in the paper. Were values from analysis of straight termites identical to that from tamandua stomachs? If not, how did the authors determine which values to use and how were they different?]

Response: The complete table with insect composition from the literature has been added to the supplementary materials.

Comment 7: [Discussion overall: The general content of the discussion seems appropriate, but some care should be given to making the concepts clearly grouped into logical paragraphs where the point being made is first introduced and then the supporting information offered below. Then some consideration to the flow would be helpful to the reader. As it is, the discussion is extremely choppy and jumps from one concept to another without clarity around the point the information being given is trying to support. At one point it appears that several lines of evidence were being offered to support a singular point, but this argument was not initially made clear to the reader, causing me to need to go back in the text several times to try to tie everything together logically, when the text itself should have done this.]

Response: Thank you for your thoughts on this. Several paragraphs have been rearranged and section headers added to make the flow clearer.

Comment 8: [Line 251: This comment about keeping vitamin K, D, and Ca within recommended ranges is arguably the most important of the paper, and yet it is not clear WHICH recommendations the authors are encouraging readers adhere to. Given the results shown in the paper, it stands to reason that current diet recommendations for this species in human care may not be appropriate and might need adjustment. Or do you mean that diet recommendations are fine as they are, but nutrient values in commercial diets do not follow these recommendations, so animal managers need to be aware? If the former, what should be the impact of this work? Further supporting already existing recommendations or suggesting adjustments that are investigated in more detail in future assessments? This is crucial information and as it stands, this main point of the work is almost lost due to a lack of clarity. Please more clearly articulate here, in the conclusion, and in the abstract WHICH recommendations are being suggested readers be mindful of.]

Response: Thank you for bringing this lack of clarity to our attention. We agree that this is a point that needs maximum clarity, and so we have adjusted the wording to reflect that the current recommendations were specifically developed with these issues of hypercalcemia in mind and are the ones to target.

Comment 9: [Lines 254-257: These appear not to belong in the discussion section and are instead instructions for writing a discussion section of a paper. Please remove them or clarify.]

Response: On our version, we do not see this. We have reviewed the document for errant instructions and have not found any. Hopefully this was a glitch in the editorial process.

Reviewer 3 Report

Comments and Suggestions for Authors

The introduction is clearly structured, describes the problem quite well and provides the reader with all the necessary information to understand the paper. I miss the effects of hypercalcemia

Collection procedure and analysed data were described quite well. I would prefer to read something about the method for analysis.

Fig.1 I guess the numbers in the dots are the numbers of specimen found dead. This should be mentioned in the description

Fig 2 and 3 boxes and lines should be explained and the order of the variables should be the same as in Table 1 and 2

I am not sure about the difference of the two tables and the two figures. Is it necessary to present both to describe the problem?

Line 206 to 224 – is a very sceptical interpretation which in my opinion is not necessary. Line 212 – is this the third evidence?

Conclusion is clear to me

Author Response

COMMENTS FROM REVIEWER 3

Comment 1: [The introduction is clearly structured, describes the problem quite well and provides the reader with all the necessary information to understand the paper. I miss the effects of hypercalcemia.]

Response: Thank you for this suggestion. We agree and have included the possible effects of hypercalcemia: “Anteaters under human care, including both giant anteaters (Myrmecophaga tridactyla) and southern tamanduas (Tamandua tetradactyla), as well as several other members of Xenarthra are frequently diagnosed with hypercalcemia, that can leads to vertebral hyperostosis, weakness, hypotonia, hyporeflexia, and cardiovascular effects such arrhythmias and hypertension [4-7].”

Comment 2: [Collection procedure and analysed data were described quite well. I would prefer to read something about the method for analysis.]

Response: Additional clarification about methods was added and a reference to the DairyOne procedures online was given.

Comment 3: [Fig.1 I guess the numbers in the dots are the numbers of specimen found dead. This should be mentioned in the description.]

Response: Thank you for your suggestion, we agree and have added to the Figure 1 description: “The specimens found are represented by black dots, numberered when nearby located”.

Comment 4: [Fig 2 and 3 boxes and lines should be explained and the order of the variables should be the same as in Table 1 and 2]

Response: Thank you for pointing out this inconsistency. The figures have been redone to increase resolution and be more consistent.

Comment 5: [I am not sure about the difference of the two tables and the two figures. Is it necessary to present both to describe the problem?]

Response: In the field of nutrition, opinions are divided on whether nutrient information should be presented on a dry matter or an energy basis. We opted to show both to appease both camps.

Comment 6: [Line 206 to 224 – is a very sceptical interpretation which in my opinion is not necessary. Line 212 – is this the third evidence?]

Response: The authors feel that the milk hypothesis is one of several strong lines of evidence for why anteaters may have this unusual metabolism and therefore require unique dietary modifications. There were a few editorial edits that may have made some sentences read poorly. These have been fixed.

Comment 7: [Conclusion is clear to me]

Response: Thank you so much for this comment.